# What are the odds? Poor compliance with UK loot box probability disclosure industry self-regulation

Leon Y. Xiao[1,2,3]*, Laura L. Henderson[3], Philip W. S. Newall[4,5]

**1** Center for Digital Play, IT University of Copenhagen, København, Denmark, **2** Department of Computer Science, University of York, York, United Kingdom, **3** The Honourable Society of Lincoln's Inn, London, United Kingdom, **4** School of Psychological Science, University of Bristol, Bristol, United Kingdom, **5** Experimental Gambling Research Laboratory, School of Health, Medical and Applied Sciences, CQUniversity, Sydney, New South Wales, Australia

* lexi@itu.dk

**Data Availability Statement:** The data presented in this paper are available via <https://doi.org/10.17605/OSF.IO/CX5RV>. Screenshots of in-game disclosures, printouts and archived links to website disclosures, and the data analysis script and output are also available at this location.

## Abstract

Loot boxes are purchased in video games to obtain randomised rewards of varying value and are thus psychologically akin to gambling. Disclosing the probabilities of obtaining loot box rewards may reduce overspending, in a similar vein to related disclosure approaches in gambling. Presently, this consumer protection measure has been adopted as law only in the People's Republic of China (PRC). In other countries, the videogaming industry has generally adopted this measure as self-regulation. However, self-regulation conflicts with commercial interests and might not maximally promote public welfare. The loot box prevalence rate amongst the 100 highest-grossing UK iPhone games was 77% in mid-2021. The compliance rate with probability disclosure industry self-regulation was only 64.0%, significantly lower than that of PRC legal regulation (95.6%). In addition, UK games generally made insufficiently prominent and difficult-to-access disclosures both in-game and on the game's official website. Significantly fewer UK games disclosed probabilities on their official websites (21.3%) when compared to 72.5% of PRC games. Only one of 75 UK games (1.3%) adopted the most prominent disclosure format of automatically displaying the probabilities on the in-game purchase page. Policymakers should demand more accountable forms of industry self-regulation or impose direct legal regulation to ensure consumer protection.

## 1. Introduction

Paid loot boxes are randomised monetisation methods in videogames that are purchased by players to obtain randomised rewards of varying value [1]. Some loot boxes may be obtained through gameplay without paying real-world money. However, the present study focuses on *paid* loot boxes which are, hereinafter, referred to as 'loot boxes.' Loot boxes are prevalent in videogames internationally and across different hardware platforms: approximately 60.0% of the highest-grossing mobile games in 'Western' countries (specifically, Australia and the UK) contain loot boxes [2, 3], as do approximately 90.0% in the People's Republic of China (PRC) [4]. (In this paper, the PRC refers to Mainland China and excludes the Special Administrative Regions of Hong Kong and Macau, and Taiwan, as the applicable laws in these areas are

**Funding:** The authors received no specific funding for this work.

**Competing interests:** L.Y.X. was employed by LiveMe, then a subsidiary of Cheetah Mobile (NYSE:CMCM), as an in-house counsel intern from July to August 2019 in Beijing, People's Republic of China. L.Y.X. was not involved with the monetisation of video games by Cheetah Mobile or its subsidiaries. L.Y.X. undertook a brief period of voluntary work experience at Wiggin LLP (Solicitors Regulation Authority (SRA) number: 420659) in London, England in August 2022. L.Y.X. has contributed and continues to contribute to research projects that were enabled by data access provided by the video game industry, specifically Unity Technologies (NYSE:U) (October 2022 – Present). L.Y.X. has met and discussed policy, regulation, and enforcement with the Belgian Gaming Commission [Belgische Kansspelcommissie] (June 2022 & February 2023), the Danish Competition and Consumer Authority [Konkurrence- og Forbrugerstyrelsen] (August 2022), the Department for Digital, Culture, Media and Sport (DCMS) of the UK Government (August 2022), PEGI (Pan-European Game Information) (January & March 2023), a member of the European Parliament (February 2023), the US Federal Trade Commission (FTC) (February 2023), and the Finnish Gambling Administration at the National Police Board [Poliisihallituksen arpajaishallinto / Polisstyrelsens lotteriförvaltning] (March 2023). L.Y.X. has been invited to provide advice to the DCMS on the technical working group for loot boxes and the Video Games Research Framework. L.Y.X. was invited to support a study on EU consumer law commissioned by the European Commission (February 2023). L.Y.X. was the recipient of two AFSG (Academic Forum for the Study of Gambling) Postgraduate Research Support Grants that were derived from 'regulatory settlements applied for socially responsible purposes' received by the UK Gambling Commission and administered by Gambling Research Exchange Ontario (GREO) (March 2022 & January 2023). L.Y.X. has accepted funding to publish academic papers open access from GREO that was received by the UK Gambling Commission as above (October, November, & December 2022). L.Y.X. has accepted conference travel and attendance grants from the Socio-Legal Studies Association (February 2022 & February 2023); the Current Advances in Gambling Research Conference Organising Committee with support from GREO (February 2022); the International Relations Office of The Jagiellonian University (Uniwersytet Jagielloński), the Polish National Agency for Academic Exchange (NAWA; Narodowa

different.) Loot boxes represent an important revenue stream for the videogame industry [5]. Certain rare rewards have a probability of as low as '0.0008%' (or 1 in 125,000) chance of being won from a loot box costing £2.50 (*e.g.*, in Game S14: *Art of Conquest*): the low entry cost and the low chance of winning, present in most loot boxes, are characteristics shared by traditional prize raffles and lotteries. Players often purchase multiple loot boxes in order to attempt to obtain the valuable rare rewards [6]. Loot boxes have been considered conceptually and structurally akin to gambling [7–9].

Vulnerable players, such as problem gamblers and children, may be at particular risk of experiencing harm when engaging with loot boxes [10–14]. In the UK, 58.9% of games deemed suitable for children aged 12+ contained loot boxes in 2019 [3], as did 90.5% in the PRC in 2020 [4]. Indeed, in 2019, 22.9% of 11–16-year olds in the UK reported paying real-world money to buy loot boxes [15], although this figure has since decreased to 10.3% in 2022 [16]. Many countries across the world are considering whether to regulate loot boxes because of their potentially harmful link to problem gambling, and because of general consumer protection concerns (*e.g.*, lack of transparency as to how the randomisation process determines loot box results) [17–19].

Not regulating loot boxes leaves consumers continually exposed to potential harms; however, banning loot boxes may be overly restrictive and unjustifiable given that only a small minority of players may be harmed [20]. Indeed, attempting to ban loot boxes may also be impractical, as demonstrated by Belgium's ineffective attempt to do so [21]. A less restrictive approach that better balances consumer freedom with consumer protection is requiring videogame companies to disclose the probabilities of obtaining randomised rewards from loot boxes, which is easy to implement and therefore incurs minimal compliance costs [4]. Such a measure seeks to provide consumers with information to help them to make more informed purchasing decisions, but does not limit consumers' ability to purchase loot boxes. Researchers have recommended adopting this measure to provide transparency and reduce the potential financial harms of overspending on loot boxes [19, 22, 23].

This probability disclosure measure has been adopted as law in the PRC, which is presently the only country to do so [4]. (Since data collection was conducted for the present study, Taiwan has also separately required this by law since July 2022 [24].) In all other countries [e.g., 25–27], the videogame industry has generally adopted this measure as a form of voluntary self-regulation or corporate social responsibility [28]: for example, all videogames published on Apple's App Store in all countries 'offering "loot boxes" or other mechanisms that provide randomized virtual items for purchase must disclose the odds of receiving each type of item to customers prior to purchase' [29].

In the PRC, amongst the 100 highest-grossing iPhone games, probability disclosures were found for 95.6% of 91 games containing loot boxes [4]. However, because the PRC law requiring probability disclosures and Apple's self-regulation were in force simultaneously, it could not be determined whether the PRC legal requirement was necessary *in addition* to Apple's self-regulation to ensure the identified high level of compliance. A replication in another country (where the relevant legal requirement does not apply, such as the UK) would shed light on the effectiveness of self-regulation acting alone.

Further, the PRC study identified a variety of different methods of probability disclosures of varying prominence and accessibility, because the relevant PRC law did not require specific methods of disclosure: only 5.5% of games disclosed probabilities in the most prominent and accessible disclosure format of automatically displaying the probabilities on the in-game loot box purchase page [4]. Probability disclosure self-regulation in Western countries is similarly worded in general terms and do not require specific, uniform and prominent methods of disclosure [30], *e.g.*, Apple's self-regulatory requirement quoted above. Corporate actions that

Agencja Wymiany Akademickiej), and the Republic of Poland (Rzeczpospolita Polska) with co-financing from the European Social Fund of the European Commission of the European Union under the Knowledge Education Development Operational Programme (May 2022); the Society for the Study of Addiction (November 2022 & March 2023); and the organisers of the 13th Nordic SNSUS (Stiftelsen Nordiska Sällskapet för Upplysning om Spelberoende; the Nordic Society Foundation for Information about Problem Gambling) Conference, which received gambling industry sponsorship (January 2023). L.Y.X. has received an honorarium from the Center for Ludomani for contributing a parent guide about a mobile game for Tjekspillet.dk, which is funded by the Danish Ministry of Health's gambling addiction pool (Sundhedsministeriets Ludomanipulje). L.Y.X. was supported by academic scholarships awarded by The Honourable Society of Lincoln's Inn (March 2020) and The City Law School, City, University of London (July 2020). L.L.H. declares no conflict of interest. P.W.S.N. is a member of the Advisory Board for Safer Gambling – an advisory group of the Gambling Commission in Great Britain, and in 2020 was a special advisor to the House of Lords Select Committee Enquiry on the Social and Economic Impact of the Gambling Industry. In the last five years P.W.S.N. has contributed to research projects funded by the Academic Forum for the Study of Gambling, Clean Up Gambling, GambleAware, Gambling Research Australia, NSW Responsible Gambling Fund, and the Victorian Responsible Gambling Foundation. P.W.S.N. has received travel and accommodation funding from the Spanish Federation of Rehabilitated Gamblers, and received open access fee grant income from Gambling Research Exchange Ontario. This does not alter our adherence to PLOS ONE policies on sharing data and materials.

seek to inhibit consumers from making more informed choices and potentially encourage them to make worse choices are termed 'sludge' in the behavioural science literature [31–34]. This contrasts with using 'nudge' to help consumers make better choices [35], which probability disclosure should do. Most disclosures observed in the PRC were arguably sludge rather than nudge because they were not uniform, not prominent, and not easily accessible, such that consumers could not derive the maximum potential benefits from them: only 21.2% of in-game disclosures and 10.6% of website disclosures in the PRC could be considered 'reasonably prominent' [4]. How games disclose probabilities in other jurisdictions governed by industry self-regulation has not been investigated. Recently, the behavioural science literature has debated whether a preoccupation with 'i-frame' interventions seeking to influence individual behaviour might have overshadowed, and made regulators less likely to consider, more systematic policy change or so-called 's-frame' interventions [36]. Information-based disclaimers are 'i-frame' interventions because they require the players, as individuals, to digest and utilise complex sets of information presented in idiosyncratic ways across games. If loot box probability disclosures as a 'i-frame' intervention is of poor efficacy globally (beyond the PRC), then countries around the world should consider more interventionist 's-frame' loot box regulation (*e.g.*, placing limits on product availability).

Finally, the only previous survey of UK loot box prevalence (which did not assess probability disclosure compliance) used a highest-grossing iPhone game list captured on 28 February 2019 [3]. It has been suggested that, since then, videogame companies have begun to stop implementing loot boxes and instead adopt other (non-randomised) monetisation methods, such as battle passes [37, 38], either to avoid bad press or to act more ethically towards their customers [39], as demonstrated by commercial decisions taken by companies such as Epic Games [40–42]. A further survey in the UK would shed light on whether loot box prevalence has indeed decreased two years after the previous UK survey.

Therefore, a survey replicating Xiao et al. (2021b) was conducted in the UK to assess: (i) the effectiveness of self-regulation alone in the absence of legal intervention; (ii) the methods of compliance (*i.e.*, prominence and accessibility of probability disclosures) in Western countries; and (iii) any industry changes in the prevalence of loot box implementation.

The following hypotheses were preregistered at <https://doi.org/10.17605/OSF.IO/FJNMY>.

Hypothesis 1: The percentage of the 100 highest-grossing iPhone games containing loot boxes in the UK that discloses loot box probabilities will be significantly lower than the 95.6% found in the highest-grossing iPhone games in the PRC by Xiao et al. (2021b).

Hypothesis 2: The percentage of the 100 highest-grossing iPhone games containing loot boxes in the UK that will be found by the present study will be significantly lower than the 59.0% found in the highest-grossing iPhone games in the UK in February 2019 by Zendle et al. (2020a).

Further, the present study was preregistered to describe:

1. the percentages of the 100 highest-grossing UK iPhone games containing loot boxes disclosing probabilities at the following three locations: (i) in-game only; (ii) on the official website only; and (iii) at both locations;

2. the percentages of the 100 highest-grossing UK iPhone games containing loot boxes using various methods of disclosure subcategories developed by Xiao et al. (2021b), and using various then yet unidentified methods of disclosure subcategories that were subsequently defined by the present study;

3. the percentage of games containing loot boxes, which were both included in Xiao et al. (2021b)'s sample and available on the UK Apple App Store with an English version when data is being collected by the present study, that disclosed loot box probabilities.

Finally, the percentage of the 100 highest-grossing UK iPhone games containing loot boxes that disclosed the implementation of pity-timer submechanics (which *change* the probabilities of obtaining randomised rewards as the player purchases more loot boxes; see [4]) is described and their prevalence rates in the UK and the PRC are compared through exploratory analysis.

## 2. Method

Replicating Xiao *et al.* (2021b) and as preregistered, the 100 highest-grossing iPhone games on the UK Apple App Store on 21 June 2021 as reported by App Annie, an authoritative independent analytics company, were selected to form the sample. In addition, as preregistered, 31 games that were both included in Xiao et al. (2021b)'s sample and available on the UK Apple App Store with an English version (but were not within the UK 100 highest-grossing list on 21 June 2021) when data was being collected by the present study were added to the sample. The aforementioned 31 games, in addition to eight games that were within both the 100 highest-grossing PRC list in Xiao *et al.* (2021b) and the UK list used by this present study, constitutes the 'Overlap Sample' of 39 games. Thus, a total of 131 games were coded. These games' titles and their numbering for the purposes of this study are shown in Table 1.

The following variables were measured:

### 2.1. Apple age rating

This variable was coded using the relevant age rating information displayed on the game's UK Apple App Store page.

### 2.2. Presence of paid loot boxes

A 'paid loot box' was defined as being either an Embedded-Isolated random reward mechanism or an Embedded-Embedded random reward mechanism, as defined by Nielsen & Grabarczyk (2019). This variable was coded based firstly on 40 minutes of gameplay. If no such mechanic was found within that time, this variable was coded based on up to 2 hours of internet browsing of video streams and screenshots. In total, 125 games (95.4%) were coded through gameplay and 6 games (4.6%) were coded through internet browsing. In contrast to Zendle et al. (2020a), but replicating Xiao et al. (2021b), games were assessed based on gameplay first, which was only then followed by internet browsing if required, because this more accurately reflected a player's experience of encountering loot boxes when they start to play a new game [4]. Additionally, this also allowed for free exploration of the game's various menus and therefore more accurate assessment of the other variables relating to probability disclosures as virtually no videos of players interacting with probability disclosures are available: videos generally only show (relatively experienced) players purchasing and opening loot boxes, without consulting the relevant probability disclosure.

### 2.3. Presence of probability disclosure

Games were coded as having disclosed probabilities if the likelihood of obtaining potential rewards from loot boxes was found either in-game or on the official website. Considerable efforts were expended when attempting to find disclosures but the risk of false negatives could not be entirely removed: however, any disclosures that were not found by the present study

**Table 1. Full list of 131 games studied and their numbering.**

| # | Title | # | Title | # | Title |
|---|---|---|---|---|---|
| 1 | Roblox | 45 | Mobile Legends: Bang Bang | 89 | Solitaire Cruise Tripeaks Card |
| 2 | Coin Master | 46 | DRAGON BALL LEGENDS | 90 | Jurassic World Alive |
| 3 | Candy Crush Saga | 47 | WWE SuperCard—Battle Cards | 91 | Clawee |
| 4 | Clash of Clans | 48 | Matchington Mansion | 92 | Football Rivals |
| 5 | PUBG MOBILE—Traverse | 49 | FIFA Soccer | 93 | Mortal Kombat |
| 6 | Clash Royale | 50 | Kiss of War | 94 | Backgammon—Lord of the Board |
| 7 | Pokémon GO | 51 | Star Trek Fleet Command | 95 | Football Manager 2021 Mobile |
| 8 | State of Survival Walking Dead | 52 | Fire Emblem Heroes | 96 | Yu-Gi-Oh! Duel Links |
| 9 | Gardenscapes | 53 | Mafia City: War of Underworld | 97 | June's Journey: Hidden Objects |
| 10 | Homescapes | 54 | Age of Z Origins | 98 | Dragon City Mobile |
| 11 | Rise of Kingdoms | 55 | Choices: Stories You Play | 99 | Golf Rival |
| 12 | Royal Match | 56 | CSR 2 Multiplayer Racing Game | 100 | Hero Wars—Fantasy World |
| 13 | Project Makeover | 57 | Empires & Puzzles Epic Match 3 | S1 | Arena of Valor |
| 14 | 8 Ball Pool | 58 | Farm Heroes Saga | S2 | LifeAfter: Night falls |
| 15 | Brawl Stars | 59 | War and Order | S3 | Princess Connect! Re: Dive |
| 16 | Golf Clash | 60 | Merge Dragons! | S4 | Arknights |
| 17 | Call of Duty: Mobile | 61 | Hay Day | S5 | Onmyoji |
| 18 | Fishdom | 62 | Family Island–Farming game | S6 | Honkai Impact 3rd |
| 19 | Bingo Blitz—BINGO games | 63 | Klondike Adventures | S7 | Shining Nikki |
| 20 | Top War: Battle Game | 64 | Lotsa Slots—Vegas Casino | S8 | Saint Seiya Awakening |
| 21 | Candy Crush Soda Saga | 65 | Manor Matters | S9 | Royal Chaos |
| 22 | Pet Master | 66 | Match Masters—PvP Match 3 | S10 | Identity V |
| 23 | Evony | 67 | Love Island The Romance Game | S11 | Last Shelter: Survival |
| 24 | Rise of Empires: Fire and War | 68 | EverMerge—Merge and Match! | S12 | SLAM DUNK |
| 25 | Toon Blast | 69 | Monster Legends: Collect all | S13 | eFootball PES 2021 |
| 26 | Minecraft | 70 | Slotomania Vegas Casino Slots | S14 | Art of Conquest |
| 27 | Genshin Impact | 71 | BitLife | S15 | Langrisser |
| 28 | Zynga Poker—Texas Holdem | 72 | Harry Potter: Puzzles & Spells | S16 | Ode To Heroes |
| 29 | Solitaire Grand Harvest | 73 | MARVEL Strike Force: Squad RPG | S17 | Azur Lane |
| 30 | Episode—Choose Your Story | 74 | Merge Mansion | S18 | Love Nikki-Dress UP Queen |
| 31 | RAID: Shadow Legends | 75 | Township | S19 | LINE: Disney Tsum Tsum |
| 32 | Lords Mobile: Tower Defence | 76 | Score! Hero 2 | S20 | BanG Dream! Girls Band Party |
| 33 | DRAGON BALL Z DOKKAN BATTLE | 77 | The Sims FreePlay | S21 | Ragnarok M: Eternal Love EU |
| 34 | Chapters: Interactive Stories | 78 | Mighty Party: Battle Heroes | S22 | Mr Love: Queen's Choice |
| 35 | Cash Frenzy—Slots Casino | 79 | The Grand Mafia | S23 | Ulala: Idle Adventure |
| 36 | Warpath | 80 | Adorable Home | S24 | Dragon Raja |
| 37 | Star Wars: Galaxy of Heroes | 81 | Garena Free Fire- World Series | S25 | PES CARD COLLECTION |
| 38 | Guns of Glory: Conquer Empires | 82 | Harry Potter: Hogwarts Mystery | S26 | Summoners War |
| 39 | Game of Thrones: Conquest | 83 | Redecor—Home Design Makeover | S27 | Sky: Children of the Light |
| 40 | Puzzles & Survival | 84 | Mario Kart Tours | S28 | Golden HoYeah Slots Casino |
| 41 | Marvel Contest of Champions | 85 | F1 Clash | S29 | Brutal Age: Horde Invasion |
| 42 | King of Avalon: Dragon Warfare | 86 | The Simpsons: Springfield | S30 | AFK Arena |
| 43 | Top Eleven Be a Soccer Manager | 87 | Last Day on Earth: Survival | S31 | Contra Returns |
| 44 | Toy Blast | 88 | MHA: The Strongest Hero | | |

were also unlikely to have been observed by and helpful to the average, or indeed even the determined, consumer.

## 2.4. Location of observed disclosure

Games were coded as having disclosed probabilities (i) in-game only, (ii) on the official website only, or (iii) at both locations. If multiple loot boxes were found for a game, this variable was coded based on all loot boxes found: *e.g.*, if a game contains loot box A and loot box B, and loot box A's probability disclosure was made in-game only and loot box B's probability disclosure was made on the official website only, this game would be coded as having made disclosures at both locations, so as to be as fair as possible to companies by giving them maximum recognition for their compliance efforts. This approach was also adopted because such potential within-game variation between multiple loot box types could not be exhaustively recorded as some games contained at least 75 different loot boxes [43].

## 2.5. Method of in-game disclosure

This variable was coded in accordance with the six subcategories of in-game disclosures developed in Xiao et al. (2021b), which included, for example, an in-game probability disclosure that is 'Automatically displayed on the loot box purchase page without requiring any additional input from the player.' When an in-game probability disclosure was found that did not fall within any of the six subcategories developed in Xiao et al. (2021b), a new subcategory was defined and created. If multiple loot boxes were found for a game, this variable was coded based on the loot box that made the most prominent in-game disclosure, so as to be fair to companies by allowing them to gain maximum credit for their compliance efforts and highlighting their most consumer-friendly examples.

## 2.6. Method of official website disclosure

This variable was coded in accordance with the five subcategories of official website disclosures developed in Xiao et al. (2021b), which included, for example, an official website probability disclosure that is 'Linked directly from the homepage.' When an official website probability disclosure was found that did not fall within any of the five subcategories developed in Xiao et al. (2021b), a new subcategory was defined and created. If multiple loot boxes were found for a game, this variable was coded based on the loot box that made the most prominent official website disclosure.

## 2.7. Was a pity-timer disclosed?

A 'pity-timer' was defined as a submechanic that changes (either increases or decreases) the probabilities of obtaining randomised rewards from loot boxes as the player purchases more loot boxes, as defined by Xiao et al. (2021b).

## 2.8. Inter-rater reliability analysis

As preregistered, 20 games (15% of the sample of 131 games, rounded up) were dual-coded to test the inter-rater reliability of the coding, which is summarised in Table 2. Two coders first coded the *Apple age rating*, *Presence of loot boxes*, *Presence of probability disclosure* and *Location of observed disclosure*. The two coders were in perfect agreement, except that there were two disagreements for the *Location of observed disclosure* (90.0% agreement, Cohen's kappa = 0.85). Discussions revealed that these related to particularly inaccessible disclosures made by Games 14: *8 Ball Pool* and S20: *BanG Dream*! *Girls Band Party*. For Game 14: *8 Ball*

**Table 2. Inter-rater reliability (n = 20).**

| Variable | Percentage agreement (Cohen's kappa) |
|---|---|
| *Apple age rating* | 100% (1.00) |
| *Presence of paid loot boxes* | 100% (1.00) |
| *Presence of probability disclosure* | 100% (1.00) |
| *Location of observed disclosure* | 90.0% (0.85) |
| *Method of in-game disclosure* | 100% (1.00) |
| *Method of official website disclosure* | 95.0% (0.93) |
| *Disclosure of a pity-timer* | 100% (1.00) |

*Pool*, which made particularly hidden disclosures at *both* locations, the first coder failed to find the in-game disclosure which could only be accessed through a button not on the loot box's purchase page (specifically, a button hidden in the settings menu that was not sign-posted from elsewhere, such as the loot box purchase screen) and the second coder failed to find the official website disclosure which was inaccessible from the official website's homepage. In addition, the first coder failed to find the official website disclosure for Game S20: *BanG Dream*! *Girls Band Party* because it could only be found on the FAQ (Frequently Asked Questions) page of the website, which, although it was linked from the homepage, did not state or imply that it would show the disclosure: the 85[th] question in a long list of 117 different questions on various topics on the FAQ page reveals how the probabilities can be accessed in-game by 'tapping the "i" on the Gacha [loot box] screen' [see 44]. Disclosing at one location how to access the disclosure at the other location was deemed to be a disclosure at both locations (if the disclosure at the other location could in fact be found as described), so as to give companies credit for at least providing these instructions to the player. The two coders then exchanged screenshots of found loot boxes, disclosures and pity-timers to ensure that both were coding based on the same loot box that made the most prominent disclosure following the methodology of Xiao et al. (2021b). There was one disagreement as to *Method of official website disclosure* (95.0% agreement, Cohen's kappa = 0.93), which was caused by the second coder creating a new subcategory to define a relatively inaccessible method of official website disclosure that was made by Game 100: *Hero Wars—Fantasy World* on the customer support website as a 'drop rates' post but which could not otherwise be accessed except through a direct link [45], unlike other customer support website disclosures which could all be found using the website's embedded search function. Following discussions, it was agreed that this proposed subcategory was redundant, and the situation was already covered by the pre-existing subcategory that the website disclosure was inaccessible from the homepage (which corresponded to the first coder's coding). The coding was adjusted after the inter-rater reliability discussions.

## 3. Results

### 3.1. Descriptive statistics: Presence of loot boxes and apple age ratings

Of the 100 highest-grossing UK iPhone games on 21 June 2021, 77.0% (77 games) contained loot boxes. Their Apple App Store age ratings are summarised in Table 3. Notably, Games 1: *Roblox* and 26: *Minecraft* were duly coded as containing loot boxes. This was because, although loot boxes were not officially implemented by the developer and publisher of these two games (*i.e.*, no 'first-party' implementation), these 'sandbox' games allow for user-generated content (UGC [46]) to be implemented and sold by third parties (which could be designed to cost real-world money and provide randomised rewards, *i.e.*, loot boxes). This is officially recognised

**Table 3. Apple App Store age rating of games containing loot boxes (cumulative; N = 100).**

| Apple App Store Age Rating | Total number of games (cumulative) | Number of games that contain loot boxes (cumulative) | Percentage containing loot boxes |
|---|---|---|---|
| 4+ | 30 | 17 | 56.7% |
| 9+ | 48 | 33 | 68.8% |
| 12+ | 80 | 61 | 76.3% |
| 17+ | 100 | 77 | 77.0% |

and permitted by Roblox Corporation [47], the developer and publisher of Game 1: *Roblox*, which explicitly requires probability disclosures for such UGC loot boxes. These two games were included in the sample when reporting *Presence of paid loot boxes* and *Apple age rating* because their existence needs to be highlighted as they present a unique compliance risk (specifically, that the subject of self-regulation may itself need to impose and enforce subsidiary self-regulation on others to ensure that it is compliant with its own self-regulation obligations, as Game 1: *Roblox* did); however, these two games were excluded from the sample when reporting presence, location and accessibility of probability disclosures and disclosure of pity-timers because the present study sought to report whether, and if so how, the game *officially* (or first-party) implemented loot box probability disclosures. The array of UGC loot boxes is too diverse and too frequently changed for these latter aspects to be assessed with confidence in these two games: doubtlessly, some UGC loot boxes do not disclose probabilities despite being required to do so, and how UGC loot boxes disclose probabilities will vary significantly between various third-party implementations.

## 3.2. Descriptive statistics: Locations of found UK disclosures

Of the 75 games containing first-party implemented loot boxes, 64.0% (48 games) disclosed probabilities as required by Apple's self-regulation, whilst 36.0% (27 games) did not. Locations at which disclosures were observed are displayed in Table 4. An exploratory binomial test revealed that the UK website disclosure availability rate of 21.3% was significantly lower ($p < .001$) than the 72.5% PRC rate [4].

## 3.3. Confirmatory analyses: Comparing disclosure and prevalence rates

The two preregistered hypotheses were tested.

Hypothesis 1 was supported using a binomial test (one-tailed test, $p = .05$) which revealed that the UK disclosure rate of 64.0% was significantly lower ($p < .001$) than the 95.6% PRC disclosure rate [4].

Hypothesis 2 was rejected using a binomial test (one-tailed test, $p = .05$) which revealed that the UK loot box prevalence rate in mid-2021 of 77.0% was not significantly lower ($p > .999$) than the 59.0% early 2019 UK prevalence rate [3]. On the contrary, it was significantly higher ($p < .001$).

**Table 4. Locations of observed disclosures (n = 75).**

| Location of Disclosure | Number of games |
|---|---|
| In-game only | 32 (42.7%) |
| On the official website only | 0 (0.0%) |
| Both locations | 16 (21.3%) |
| No disclosure found | 27 (36.0%) |

## 3.4. Descriptive statistics: Accessibility of UK in-game and website disclosures

Eight subcategories of in-game disclosures were identified, as summarised in Table 5. The UK situation was similar to the PRC's [4]: more than half of games' probability disclosures in both jurisdictions were accessed through tapping a small generic button. Three new subcategories, not identified in the PRC [4], were defined: although they were each only subtly different from pre-existing subcategories, certain aspects of their implementation nonetheless render them distinct and noteworthy. For example, two games disclosed probabilities when symbols that conceptually implied randomness and chance (*e.g.*, a dice symbol used in Game 87: *Last Day on Earth*: *Survival*) were interacted with: this subcategory therefore should be recognised as being different from the subcategory that showed probabilities after a small generic symbol, such as an '*i*' or '(?)' button (which do not in any way allude to probabilities, as used in Game S14: *Art of Conquest*) was tapped (26 games). Both aforementioned subcategories should be deemed less prominent than the subcategory that displayed probabilities after a button explicitly stating 'probabilities' or a conceptually similar word (*e.g.*, 'rates' used in Game 33: *DRAGON BALL Z DOKKAN BATTLE*) was tapped (3 games). One egregiously hidden in-game disclosure subcategory (that was initially missed by a coder as described in the Method section) was used by one game (Game 14: *8 Ball Pool*): players were required to enter the settings menu, scroll down to the bottom and then interact with a button to 'View' 'Mini Games Information' in order to be redirected to the website disclosure This arguably actively concealed implementation draws some allusions to the one PRC game that required players to

**Table 5. Subcategories of observed in-game disclosures (n = 48).**

| Number of games | Adoption rate | Summary of disclosure format | Further implementation details |
|---|---|---|---|
| 26 (54.2%) | 34.7% | Immediately after tapping a small generic symbol | *e.g.*, a question mark sign '(?)' (Game S14: *Art of Conquest*); an 'i' or '*i*' sign, which stands for 'information' (Game 31: *RAID*: *Shadow Legends*); an exclamation mark sign ['!'] (Game S23: *Ulala*: *Idle Adventure*); or a 'details' button (Game 27: *Genshin Impact*) |
| 13 (27.1%) | 17.3% | After tapping a small generic symbol and following additional steps | Same types of generic symbol as above. Additional steps include, *e.g.*, tapping on another button (Game 77: *The Sims FreePlay*); or tapping on another button and following a hyperlink to the game's official website's disclosures (Game 6: *Clash Royale*) |
| 3 (6.3%) | 4.0% | Immediately after tapping a small button explicitly referencing 'probabilities' or a conceptually similar term | *e.g.*, a button stating 'Character Summoning Rates' (Game 33: *DRAGON BALL Z DOKKAN BATTLE*); 'Appearance Rates' (Game 52: *Fire Emblem Heroes*); or 'Drop Rate' (Game S22: *Mr Love*: *Queen's Choice*) |
| 2 (4.2%) | 2.7% | Interacting with a graphic symbol that conceptually referenced 'probabilities' and 'chance' | *e.g.*, a dice symbol (Game 87: *Last Day on Earth*: *Survival*) |
| 1 (2.1%) | 1.3% | Automatically displayed on the loot box purchase page without requiring any additional input from the player | Specifically, as implemented in Game 98: *Dragon City Mobile* |
| 1 (2.1%) | 1.3% | After tapping a small button explicitly referencing 'probabilities' and following additional steps | Specifically, tapping a 'Pack Probabilities' hyperlink button and then tapping a 'Continue' button that takes the player to the official website disclosure (Game 37: *Star Wars*: *Galaxy of Heroes*) |
| 1 (2.1%) | 1.3% | By tapping a graphic element on the loot box purchase page that was not seemingly interactable and then following additional steps | Specifically, tapping the picture depicting the loot box above the payment/price button (colloquially known to players as the loot box 'banner') and then tapping an [*i*] button and an 'OK' button to confirm being redirected to the official website disclosure (Game 69: *Monster Legends*) |
| 1 (2.1%) | 1.3% | By interacting with certain buttons not on the loot box purchase page | *e.g.*, a button hidden within the game's settings menu (Game 14: *8 Ball Pool*) |

*Note*. Adoption rate refers to the percentage of the 75 games implementing first-party loot boxes that adopted each subcategory. Example games used to illustrate each subcategory were not necessarily included in the subsample.

chat, in a foreign language, with the in-game customer support bot found in the settings menu in order to access the disclosure [4]. In contrast, only one game (Game 98: *Dragon City Mobile*) adopted the most prominent disclosure method of automatically showing the probabilities on the loot box purchase page without requiring any additional input from the player.

Four subcategories of website disclosures were identified, as summarised in Table 6. The much smaller subsample size of 16 games, when compared to the PRC's of 66 games [4], caused by companies not disclosing probabilities on official websites in the UK must be noted. The results therefore need to be interpreted with some caution. Unlike with in-game disclosures, the UK website disclosure situation is very different from the PRC situation [4]: the vast majority of PRC website disclosures (78.8% in the PRC) were published as 'news' or 'notice' posts and then chronologically listed alongside other posts; in contrast, only two UK games (12.5%) disclosed probabilities on their websites in this manner. Two new subcategories, not previously identified in the PRC [4], were defined. Firstly, two games' website disclosures were technically linked from their respective homepages; however, the interactable link on the homepages did not explicitly reference 'probabilities' or in any way allude to the disclosure being available on the other page that will be opened [see 44]. Secondly, nearly half of UK website disclosures (7 games) were published as 'probabilities' or 'drop rates' posts on the customer support website and could be found using the website's embedded search function but required players to perform this additional step. Finally, nearly a third of UK website disclosures (5 games) were not accessible from the website's homepage and could only be accessed either through typing in and visiting the correct URL, or through being redirected from in-game: these were technically official website disclosures, but were effectively inaccessible by people who are unfamiliar with, or do not play, the game (*e.g.*, parents of child players).

## 3.5. Descriptive statistics: Overlap sample

Of the Overlap Sample of 39 games whose disclosure status were assessed in both the PRC by Xiao et al. (2021b) and the UK by the present study, 13 games (33.3%) had different Apple age ratings, of which four games (30.8%) had higher age ratings in the UK and nine games (69.2%) had higher ratings in the PRC.

**Table 6. Subcategories of observed website disclosures (n = 16).**

| Number of games | Adoption rate | Summary of disclosure format, including link to example implementation |
|---|---|---|
| 7 (43.8%) | 9.3% | Published as a 'probabilities' or 'drop rates' post on the customer support website and could be found using the website's search function, *e.g.*, Game 15: *Brawl Stars* [57] |
| 5 (31.3%) | 6.7% | Inaccessible from the homepage (*i.e.*, a web address exists for the disclosure, but the link can only be found through a search engine or is only linked to from in-game, such that the disclosure on the official website is not hyperlinked from anywhere else on the website), *e.g.*, Game 37: *Star Wars*: *Galaxy of Heroes* [43] |
| 2 (12.5%) | 2.7% | Published under the 'news' or 'notice' tab and which were then chronologically listed alongside other posts, *e.g.*, Game 6: *Clash Royale* [62] |
| 2 (12.5%) | 2.7% | Published on a page that is linked directly from the homepage; however, the link does not reference 'probabilities' or 'drop rates' and therefore it is unclear that the link leads to the disclosure, *e.g.*, for Game S20: *BanG Dream*! *Girls Band Party*, on the FAQ page of the website as described in the Method section [44] |

*Note*. Adoption rate refers to the percentage of the 75 games implementing first-party loot boxes that adopted each subcategory. Example games used to illustrate each subcategory were not necessarily included in the subsample.

Loot boxes were found in the same 35 games (89.7%) in both jurisdictions. The disclosure rate was identical in both jurisdictions at 94.3% (33 of 35 games). However, there were four inconsistencies (11.4%) as to whether a game disclosed probabilities: two games' disclosures were found only in the PRC [4], whilst two other games' disclosure were found only in the UK by the present study. These inconsistencies were not further investigated because they may have been caused simply by the passage of time.

A binomial test revealed that the disclosure rate of 94.3% in the Overlap Sample is significantly higher ($p < .001$) than the 64.3% found in the overall sample when overlapping games were excluded ($n = 70$).

### 3.6. Exploratory analyses

**3.6.1. Comparisons with PRC loot box prevalence rates.** A binomial test revealed that the UK loot box prevalence rate in mid-2021 of 77.0% found by the present study remained significantly lower ($p < .001$) than the 91.0% mid-2020 PRC prevalence rate [4]. Of games deemed suitable for children aged 12+, binomial tests revealed that the loot box prevalence rate of 76.3% found by the present study was significantly higher ($p < .001$) than the 58.9% early 2019 UK prevalence rate [3], and significantly lower ($p < .001$) than the 90.5% mid-2020 PRC prevalence rate [4].

**3.6.2. Disclosures are less likely to be found for UK-only games.** Of the 100 highest-grossing UK iPhone games on 21 June 2021, 73% (73 games) were not available in the PRC Apple App Store when a search was conducted on 11 September 2021. (This search was conducted using best endeavours; however, it may potentially have incorrectly categorised certain games as being available in the UK only because, for example, certain games might have initially been made available on the PRC store but were removed from the PRC store by 11 September 2021. The results under this subheading should therefore be interpreted with due caution; however, note that such one direction errors could only render the probability disclosure compliance rate in the UK only subsample to be higher than the true value, meaning that the interpretations made by the present study hold even if errors along these lines were made.) Of these 73 games, 76.7% (56 games) contained loot boxes, of which, 60.7% (34 games) disclosed probabilities. Amongst the 105 games containing first-party implemented loot boxes in all 131 games sampled, a two-sample z-test revealed that probability disclosures were significantly less likely to have been found for games available in the UK only (of which 60.7% made disclosures) when compared to games available in both jurisdictions (of which 89.8% made disclosures), $z = 3.40$, $p < .001$.

**3.6.3. Pity-timers.** Of 75 games containing first-party implemented loot boxes within the UK 100 highest-grossing list, 34.7% (26 games) disclosed the implementation of a pity-timer mechanic. A binomial test revealed that this disclosed UK pity-timer prevalence rate was significantly lower ($p < .001$) than the 65.9% mid-2020 PRC pity-timer prevalence rate [4].

## 4. Discussion

### 4.1. Loot box prevalence and accessibility to children

In-game purchases involving randomisation were prevalently implemented in 77.0% of the highest-grossing iPhones games available on the UK Apple App Store. Hypothesis 2 that loot box prevalence in the UK has decreased in the two years following the previous UK study due to industry developments (*e.g.*, abandonment of loot box implementation for ethical reasons) was rejected. It would appear that corporate actions by particular companies like Epic were special cases that have been overemphasised and do not reflect a broader trend within the industry, at least on the mobile market. Most high-grossing companies did not stop

implementing loot boxes in order to provide better consumer protection. A significantly higher UK loot box prevalence rate than Zendle et al.'s 2019 results (59.0%) (2020a) was found. Discussions between the research teams revealed that this is partially due to: (i) more games implementing loot boxes; (ii) Zendle et al. failing to find the more minor and hidden loot boxes in a few games (similar to those found in Game 53: *Mafia City*: *War of Underworld*); and (iii) Zendle et al. not recognising social casino games (wherein the player is able to spend real-world money to participate in randomised gambling activities but cannot withdraw any winnings into cash) as containing 'loot boxes,' contrary to the present study's methodology [48, cf. 49]. Notably, the true prevalence rate of loot boxes could actually be higher than the reported 77.0% because it is possible for loot box implementations to have been missed by both coders, but it is *not* possible for any game to have been mistakenly identified as containing loot boxes when it did not (because screenshots of all identified loot boxes are available for public scrutiny at the data deposit link: <https://doi.org/10.17605/OSF.IO/CX5RV>). In addition, loot boxes were more prevalent within each age rating category than Zendle et al. (2020a) had suggested: for example, 76.3% of games deemed suitable for children aged 12+ contained loot boxes in 2021, compared to the 58.9% in 2019. Loot boxes are more widely available and easily accessible to children and young people in the UK than was previously apparent.

A number of differences were identified between the PRC and UK samples: the 77.0% loot box prevalence rate remained significantly lower than that found in the PRC in 2020 (91.0%); pity-timers were disclosed significantly less prevalently in the UK (34.7%) as compared to the PRC (65.9%) (although the actual prevalence of pity-timers was not measured); and 33.3% of games in the Overlap Sample had different age ratings. The product availability in various countries can be vastly different: surveys should be conducted in other countries to assess national loot box prevalence before any regulations are imposed. However, the present results are highly likely to be translatable internationally to other Western countries in North America, Europe, and Australasia, where the highest-grossing game lists are similar.

## 4.2. Ineffectiveness of industry self-regulation

Probability disclosures, as required by Apple's self-regulation, were found for only 64.0% of the 75 games within the UK 100 highest-grossing list containing first-party implemented loot boxes. Considerable efforts (above and beyond what a regular consumer might have used) were expended to try to find the disclosures: any that were not found were also unlikely to have been found by the average, or even the determined, player. Indeed, games whose disclosures could not be found highly likely *failed* to disclose probabilities as required [4]. The following discussion proceeds on that assumption.

Players of 36.0% of games containing first-party implemented loot boxes presumably had no access to probability disclosures, despite this consumer protection measure having been promised to them by both Apple and the relevant game companies. Apple is the owner and operator of the software marketplace, and therefore it financially benefits from every loot box sale because it always deducts a commission of up to 30% and at least 15%. Apple is also the self-regulator that sets out the relevant loot box probability disclosure requirement. However, Apple has not sought to actively enforce its self-regulation and police compliance; on the contrary, Apple is arguably financially benefiting from the lack of probability disclosures in more than a third of games containing loot boxes when players potentially overspend: there is a conflict of interest in that the rule-making self-regulator benefits from non-compliance. The games analysed by the present study were the most popular and highest-grossing UK iPhone games (the compliance rate amongst worse-performing games is likely even lower), and therefore Apple's inaction with this particular sample could not be excused for practical reasons,

such as being unable to assess every single one of more than one million games, reportedly presently available for download from the Apple App Store [50]. Indeed, Apple is arguably obliged to audit compliance amongst all available games, in accordance with the promise it made to consumers when it instituted the self-regulation. Regardless, Apple can afford to, and is obliged to, monitor compliance and rectify non-compliance amongst *at least* the highest-grossing games, but Apple has *failed* to do so.

The relevant game companies that did not make loot box probability disclosures are failing to be transparent and honest not only with players but also with Apple, because companies intending to market games on the App Store must purport to have complied with the App Store Review Guidelines, which includes the loot box probability disclosure self-regulation [29], during the submission review process [51]. Players of a substantial proportion of games are not being given the consumer protection that was promised to them. Crucially, such non-compliance with self-regulation that deceptively purported to have been compliant has given consumers, and also policymakers and regulators, the *false* impression that industry-wide compliance (and a certain degree of consumer protection) has been achieved in Western countries, despite loot box probability disclosures not being required by law when this is evidently untrue. Failing to provide disclosures after promising to provide them is arguably worse than not promising and not providing disclosures in the first place because, in the latter case, consumers would know that they are not being provided with any consumer protection and therefore may act in a more risk-averse manner, whilst in the former case, consumers were misled into believing that consumer protection is being provided to them (when it is not) and therefore they may have been induced to act in a more risk-tolerant manner by this misrepresentation.

European consumer protection law certainly recognises that the former is *worse* and more culpable than the latter: for example, 'claiming to be the signatory to a code of conduct [*i.e.*, self-regulation] when the trader is not,' or 'claiming that a trader (including his commercial practices [*i.e.*, whether or not to disclose loot box probabilities]) or a product has been approved, endorsed or authorised by . . . a private body [*i.e.*, an industry self-regulator such as Apple] when the trader, the commercial practices or the product have not or making such a claim without complying with the terms of the approval, endorsement or authorisation' are both viewed as 'commercial practices which are in all circumstances considered unfair' that are liable for criminal prosecution (EU Unfair Commercial Practices Directive [2005] OJ L149/22, Annex 1, paras 1 and 4). An example to illustrate this may be that parents would be more willing to allow their children to play games containing loot boxes if the parents know that the game makes probability disclosures: thus, a game which purports to, but does not, disclose probabilities is potentially more harmful because it creates a false impression of being 'safer' and enforcement action against it is more imminently necessary.

Companies were statistically significantly more likely to disclose probabilities in the PRC where legal requirements applied than they were to disclose in the UK where only advisory-level industry self-regulation applied. This proposition is further supported by the finding that games available in the UK only were statistically significantly less likely to have made disclosures (only 60.7% did so) than games available in both the UK and the PRC (89.8% did so), and the particularly high disclosure rate in the Overlap Sample (which consisted of well-performing and highly scrutinised PRC games that were also available in the UK) of 94.3%. A spillover effect may have been observed: companies intending to release games in both the UK and the PRC markets simply made a PRC-law compliant version of the game and released said version in all markets (regardless of the absence of local legal requirements and/or industry self-regulation). The PRC legal regulation may have improved the degree of consumer protection in other countries beyond its formal jurisdiction.

Policymakers and regulators in countries such as the UK, where practically voluntary and non-enforced industry self-regulation similar to Apple's is already in force, should demand more accountable forms of industry self-regulation and consider requiring loot box probability disclosure by law to increase the rate of compliance and better protect consumers from potential loot box harms, *e.g.*, overspending.

## 4.3. Locations and methods of disclosure

Similarly to the pattern identified in the PRC [4], companies preferred to make disclosures at only one location (66.7%), rather than at both locations (33.3%). This means that, in relation to most games, players do not have two alternative channels of accessing probability disclosures, thereby reducing the number of players seeing the disclosures and therefore providing a lower degree of consumer protection. Notably, unlike in the PRC where companies preferred making website disclosures, companies in the UK preferred making in-game disclosures. No game in the UK disclosed on the official website *only* (0.0%), whilst 40.2% did so in the PRC. Indeed, in total, 75.9% of games containing loot boxes (66 of 87) made disclosures on the official website in the PRC, but only 33.3% did in the UK (16 of 48 games). This is a statistically significant difference between the UK and the PRC. Survey results have shown that in-game disclosures are more likely to be seen by players than website disclosures, even in the PRC where website disclosures are implemented more frequently than in-game disclosures [52]. A preference for in-game disclosures may be beneficial for consumer protection; however, making disclosures at both locations is evidently superior: on one hand, making disclosures internally within the game ensures that the disclosures are more proximate to the loot box purchase decision itself, which likely maximises their effect at providing a 'cooling off' period and potentially halting purchase [53, 54]; on the other hand, making disclosures externally on the official website ensures that non-players who have an interest in learning about the disclosures (*e.g.*, parents of child players) can also easily access them without needing to expend time to play the game themselves [30]. However, two-thirds of UK games *failed* to make disclosures at both locations.

Xiao et al. (2020b) defined a disclosure as being 'reasonably prominent' if the player can access it by interacting with an element either on the loot box purchase page or the website's homepage that referenced 'probabilities' or a conceptually similar term (*e.g.*, 'rates'). Of all UK in-game disclosures, only five games' disclosures (10.4% of in-game disclosures) could be deemed reasonably prominent because they were either shown automatically (one game), or were accessed through initially interacting with a button on the loot box purchase page that explicitly referenced 'probabilities' or 'rates' so as to clearly indicate that the button led to the disclosure (three games showed the disclosure immediately upon the button being tapped, whilst one other game required performing additional steps after tapping said button). Importantly, when compared, UK games made fewer reasonably prominent in-game disclosures (10.4%) than in the PRC (21.2%).

None of the UK website disclosures (0.0%) could be deemed 'reasonably prominent' because the player could not access any of them by simply interacting with an automatically displayed element on the website's home page that referenced 'probabilities' or a conceptually similar term. One UK website disclosure subcategory representing 43.8% of website disclosures was comparatively more accessible than the other subcategories because players could access the disclosures by using the customer support website's homepage's embedded search function to look up terms such as 'probability' (Game 2: *Coin Master* [55]), 'chances' (Game 56: *CSR 2 Multiplayer Racing Game* [56]), or 'drop rates' (Game 15: *Brawl Stars* [57]). Notably, the terms generally were not interchangeable, meaning that, for example, if a player searched

for 'probability' on the official website of Game 15: *Brawl Stars*, there would have been no result and the player would not have been able to access the disclosure. Companies could have inputted all these synonyms as keywords on the probability disclosure page to ensure that it would be found when any one of the synonyms was searched for by the player. This subcategory could not be deemed reasonably prominent because it required the player to perform multiple additional steps (opening up the embedded search function; typing in the correct keyword search phrase; and then following the hyperlink that is shown amongst the search results), and because there was a chance that the player might enter an incorrect synonym and believe that there was no disclosure available. When compared, fewer official website disclosures were reasonably prominent in the UK (0.0%) than in the PRC (10.6%). Substantially more UK website disclosures were also entirely inaccessible from the homepage (31.3%) than in the PRC (7.6%).

Overall, of all 75 highest-grossing UK iPhone games containing first-party implemented loot boxes, only one game (1.3%) adopted the most prominent disclosure format of automatically displaying the probabilities on the in-game loot box purchase page (compared to 4 of 91 PRC games (4.4%)); only five UK games (6.7%) made reasonably prominent in-game disclosures (compared to 11 of 91 PRC games (12.1%)); and no UK games (0.0%) made reasonably prominent official website disclosures (compared to 7 of 91 PRC games (7.7%)). Overall, only 5 UK games (6.7%) made reasonably prominent disclosures at either location (compared to 17 of 91 PRC games (18.7%)) and no game (0.0%) made reasonably prominent disclosures at both locations (compared to 1 of 91 PRC games (1.1%)). When compared to PRC games, UK games generally made less prominent and less accessible disclosures both in-game and on the official website, and significantly fewer UK games disclosed probabilities on their official websites at all. In terms of the method of disclosure, the UK and the PRC loot box probability disclosure compliance situation appears similarly suboptimal: nearly *all* companies in both countries failed to adopt the most prominent disclosure format of automatically displaying probabilities in-game and failed to make reasonably prominent disclosures at both locations.

## 4.4. A preponderance of sludge

Sludge (*i.e.*, ways that information is obscured or complicated by companies to inhibit informed consumer choice) was widely deployed by videogame companies in the UK loot box probability disclosure context. The present study was unable to verify the accuracies of disclosures made due to lack of access to previous opening results held by game companies. However, examples of evidently unclear, inaccurate, or incomplete probability disclosures were identified: for example, Game 77: *The Sims FreePlay* disclosed percentages that did not sum to 100%. Game S31: *Contra Returns* disclosed the probability of obtaining many specific loot box rewards not as a percentage but instead as a range, *e.g.*, '0.5%–2%' and '45%–70%,' without any further explanation.

Sludge can also be seen in the differing methods of disclosure: for example, with in-game disclosures, there was great variability in what button on the purchase page the player must first engage with to access the disclosure or at least to begin the process of eventually accessing the disclosure (compare the '(?)' button used in Game S14: *Art of Conquest*, the dice symbol used in Game 87: *Last Day on Earth: Survival*, the 'Character Summon Rates' button used in Game 33: *DRAGON BALL Z DOKKAN BATTLE*, and the '(*i*)' button used in Game 77: *The Sims FreePlay*). Confusingly, certain games chose to use such symbols, which would have led to disclosures in other games, for other purposes unrelated to probability disclosures: for example, in Game 8: *State of Survival Walking Dead*, the '(i)' button on the loot box purchase screen, when tapped, displayed a reward preview screen, which easily could have, but

importantly did *not*, display probability disclosures. In most cases, players would not automatically know which button they should interact with to access the disclosure and would have to try multiple options, because there is no uniform industry standard requiring a specific format, unlike standardised food nutrition labels, for example [58]. A button explicitly stating 'probability disclosures' would be the most easily recognisable in all circumstances, but only 6.6% of games making in-game disclosures adopted this or automatically displayed probabilities.

### 4.5. How can companies do better?

The methods of disclosure, both in-game and on the official websites, and the examples of sludge presented by this study echo the suboptimal compliance situation identified in relation to loot box probability disclosure law in the PRC [4]. Companies chose to comply with loot box probability disclosure self-regulation imposed by Apple in 64.0% of games containing loot boxes; however, most companies that have disclosed probabilities used methods of disclosure that are difficult for players to access, and many companies have adopted corporate actions that obfuscate and complicate the decision-making environment and further discourage players from seeing the disclosure and benefiting from their consumer protection effects.

Companies should make disclosures both in-game on the loot box purchase page and on the game's official website to maximise players' and non-players' opportunities to access and benefit from the probabilities. Companies can easily improve the prominence and ease of access of their disclosures by automatically showing them on the in-game loot box purchase page or, if that is not possible, by adopting the other reasonably prominent disclosure methods recognised above under Section 4.3.. Companies should also consider whether their disclosures should list each individual reward separately, or disclose probabilities per rarity category and have a separate list of rewards in each category [see 30].

### 4.6. From individual-based intervention to systematic change?

Indeed, to place this case study on video game loot box probability disclosures into the wider behavioural science literature, probability disclosures represent an 'i-frame' intervention that seeks to influence the individual's behaviour for the better (but ultimately places the burden on the individual to change, with some assistance) [36]. The industry obviously supports this measure because it maintains the status quo and its positive effects are likely *de minimis*. Other 's-frame' interventions that seek to change the underlying system are needed in the loot box context to achieve better consumer protection on a wider scale, *e.g.*, how loot boxes are designed and how video games are monetised. Companies should, of course, refrain from using sludge to weaken the effectiveness of probability disclosures (thus improving the effectiveness of the i-frame intervention). However, more importantly, systematic changes (*i.e.*, s-frame interventions) should encourage, if not require, companies to design more 'ethical' and safer video game monetisation methods: for example, by forcing companies to implement fewer different loot box types in each game and fewer potential rewards in each loot box. Doing so would reduce the length and enhance the readability of the probability disclosure and possibly even eliminate the need of such an in-frame intervention entirely [59]. The mere existence of probability disclosures as an i-frame intervention should not dissuade policy-makers from seeking more systematic changes (*i.e.*, s-frame interventions), such as banning loot boxes for under-18s, if better consumer protection on a wider scale is deemed desirable.

### 4.7. Limitations

The present study focused on the highest-grossing games that were the most scrutinised by the public: this likely means that the compliance rate would have been lower amongst financially

worse-performing games that received less public oversight. The compliance situations on Android, the other major mobile platform, and other hardware platforms (*i.e.*, amongst console and PC games) remain to be specifically assessed by future research. In particular, a future study should consider the compliance situation on the Google Play Store for Android [see 60]: this is because Apple iOS devices are generally more expensive than Android devices meaning that the two platforms might be used by different players and companies, such that the present iOS-based results may have been affected by socioeconomic status bias and therefore not replicate to the Android platform. Cultural differences between the UK and the PRC might have partly contributed to the observed marked differences in compliance: specifically, if the UK does adopt loot box probability disclosure regulation as law, the compliance rate might increase but still remain lower than that previously observed in the PRC.

The present study did not assess whether more established companies or member companies of specific trade associations, *e.g.*, the ESA or Ukie, have complied with the self-regulation more widely and more effectively than other companies. Note, however, that in light of the present results, even if all members of a trade syndicate have complied with the industry self-regulation, that syndicate should be cautious when communicating to the public about the transparency and safety of the product. This is because that syndicate has no control over, and cannot regulate, the compliance (or lack thereof) of non-syndicate member companies. Widespread industry compliance is not present; therefore, such a false sense of security should not be impressed upon consumers.

The relevant features of only one loot box were recorded per game. Most games contained multiple loot box types: for example, Game 37: *Star Wars*: *Galaxy of Heroes*'s website disclosure revealed that it contained at least 75 different loot box types [43]. Exhaustively recording the relevant features of each loot box type in each game was not practicable. Indeed, variations in loot box implementation within the same game could not be recorded: each game was coded according to the loot box that used the most prominent disclosure method, even though in some games other loot box types disclosed probabilities using worse methods, for example, in Game 33: *DRAGON BALL Z DOKKAN BATTLE*, one loot box's in-game disclosure was accessed by tapping a button stating 'Character Summon Rates,' whilst another loot box's in-game disclosure was accessed by tapping an identical button stating 'Character Summon Rates' but then tapping another button on a new screen that opens up: this within-game variation demonstrates that individual players may experience loot box probability disclosures differently even within the same game.

Whether probability disclosures are effective at reducing loot box spending or preventing overspending remains to be assessed through further research. Survey results from PRC players indicate that only 16.4% of players self-reported spending less money on loot boxes after seeing probability disclosures [52]. Individual players' loot box spending data from before and after probability disclosures were implemented would reveal the measure's effectiveness: videogame companies are in possession of such data and should share them publicly to ensure that players are provided with adequate consumer protection measures, rather than an ineffective measure that is merely performative. Probability disclosures are not the only form of harm-minimisation technique that might be potentially beneficial and could be adopted: other options such as capping the amount of money that players can spend [53, 61] or reducing the complexity, and other potentially problematic aspects, of loot boxes [22, 39, 59] should also be considered.

## 5. Conclusion

Loot boxes are now more prevalent in UK videogames and more accessible to UK children than previously suggested. Amongst the highest-grossing UK iPhone games, industry self-

regulation requiring probability disclosures has resulted in only a 64.0% compliance rate, significantly lower than the 95.6% disclosure rate in the PRC where loot box probability disclosures are required by law. Emerging technologies are often initially subject only to industry self-regulation. The public and policymakers should, justifiably, be sceptical of the potential effectiveness of any proposed voluntary self-regulation with little enforceability and no independent oversight, and be wary of the motivations of the industries recommending self-regulation for adoption in lieu of legal regulation. Policymakers around the world should demand more accountable forms of self-regulation and, if that is not forthcoming, require loot box probability disclosure by law to ensure a higher compliance rate and provide better protection to consumers.

When given discretion as to how prominently and accessibly probability disclosures should be implemented, the vast majority of companies chose methods that are suboptimal: for example, by failing to disclose probabilities at multiple alternative locations; by requiring players to perform complex series of actions before showing them the disclosures; and by disclosing probabilities using technical language and mathematical formulae (as was done by Game 100: *Hero Wars–Fantasy World*) that are difficult to understand. The videogame industry, both companies and self-regulators (*e.g.*, software marketplaces and hardware providers), can do much better when it comes to making and requiring uniform, prominent and accessible loot box probability disclosures. Policymakers should not treat requiring probability disclosures as an adequate regulatory solution to the potential harms of loot boxes. More could be done.

## Author Contributions

**Conceptualization:** Leon Y. Xiao, Laura L. Henderson, Philip W. S. Newall.

**Data curation:** Leon Y. Xiao.

**Formal analysis:** Leon Y. Xiao.

**Investigation:** Leon Y. Xiao, Laura L. Henderson.

**Methodology:** Leon Y. Xiao, Laura L. Henderson.

**Project administration:** Leon Y. Xiao.

**Resources:** Leon Y. Xiao.

**Software:** Leon Y. Xiao.

**Supervision:** Philip W. S. Newall.

**Validation:** Leon Y. Xiao, Laura L. Henderson.

**Visualization:** Leon Y. Xiao.

**Writing – original draft:** Leon Y. Xiao, Laura L. Henderson, Philip W. S. Newall.

**Writing – review & editing:** Leon Y. Xiao, Laura L. Henderson, Philip W. S. Newall.

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
