## [Decision Letter · Decision Letter 0]

3 Apr 2023

PONE-D-23-02587What are the odds? Poor compliance with UK loot box probability disclosure industry self-regulationPLOS ONE

Dear Dr. Xiao,

Thank you for submitting your manuscript to PLOS ONE. After careful consideration, we feel that it has merit but does not fully meet PLOS ONE’s publication criteria as it currently stands. Therefore, we invite you to submit a revised version of the manuscript that addresses the points raised during the review process.

We look forward to receiving your revised manuscript.

Kind regards,

Simon Grima, PhD

Academic Editor

PLOS ONE

“L.Y.X. was employed by LiveMe, then a subsidiary of Cheetah Mobile (NYSE:CMCM), as an in-house counsel intern from July to August 2019 in Beijing, People’s Republic of China. L.Y.X. was not involved with the monetisation of video games by Cheetah Mobile or its subsidiaries. L.Y.X. undertook a brief period of voluntary work experience at Wiggin LLP (Solicitors Regulation Authority (SRA) number: 420659) in London, England in August 2022. L.Y.X. has met and discussed policy, regulation, and enforcement with the Belgian Gaming Commission [Belgische Kansspelcommissie] (June 2022), the Danish Competition and Consumer Authority [Konkurrence- og Forbrugerstyrelsen] (August 2022) and the Department for Digital, Culture, Media and Sport (DCMS) of the UK Government (August 2022). L.Y.X. has been invited to provide advice to the DCMS on the technical working group for loot boxes and the Video Games Research Framework. L.Y.X. was the recipient of two AFSG (Academic Forum for the Study of Gambling) Postgraduate Research Support Grants that were derived from ‘regulatory settlements applied for socially responsible purposes’ received by the UK Gambling Commission and administered by Gambling Research Exchange Ontario (GREO) (March 2022 and January 2023). L.Y.X. has accepted funding to publish academic papers open access from GREO that was received by the UK Gambling Commission as above (October, November, and December 2022). L.Y.X. has accepted conference travel and attendance grants from the Socio-Legal Studies Association (February 2022), the Current Advances in Gambling Research Conference Organising Committee with support from GREO (February 2022), the International Relations Office of The Jagiellonian University (Uniwersytet Jagielloński), the Polish National Agency for Academic Exchange (NAWA; Narodowa Agencja Wymiany Akademickiej) and the Republic of Poland (Rzeczpospolita Polska) with co-financing from the European Social Fund of the European Commission of the European Union under the Knowledge Education Development Operational Programme (May 2022), and the Society for the Study of Addiction (November 2022). L.Y.X. was supported by academic scholarships awarded by The Honourable Society of Lincoln’s Inn and The City Law School, City, University of London.

L.L.H. declares no conflict of interest.

P.W.S.N. is a member of the Advisory Board for Safer Gambling – an advisory group of the Gambling Commission in Great Britain, and in 2020 was a special advisor to the House of Lords Select Committee Enquiry on the Social and Economic Impact of the Gambling Industry. In the last five years P.W.S.N. has contributed to research projects funded by the Academic Forum for the Study of Gambling, Clean Up Gambling, GambleAware, Gambling Research Australia, NSW Responsible Gambling Fund, and the Victorian Responsible Gambling Foundation. P.W.S.N. has received travel and accommodation funding from the Spanish Federation of Rehabilitated Gamblers, and received open access fee grant income from Gambling Research Exchange Ontario.”

4. We note that 1,2,3,4,5,6,7,8 and 9 in your submission contain copyrighted images. All PLOS content is published under the Creative Commons Attribution License (CC BY 4.0), which means that the manuscript, images, and Supporting Information files will be freely available online, and any third party is permitted to access, download, copy, distribute, and use these materials in any way, even commercially, with proper attribution. For more information, see our copyright guidelines: http://journals.plos.org/plosone/s/licenses-and-copyright.

a. You may seek permission from the original copyright holder of 1,2,3,4,5,6,7,8 and 9 to publish the content specifically under the CC BY 4.0 license.

b.If you are unable to obtain permission from the original copyright holder to publish these figures under the CC BY 4.0 license or if the copyright holder’s requirements are incompatible with the CC BY 4.0 license, please either i) remove the figure or ii) supply a replacement figure that complies with the CC BY 4.0 license. Please check copyright information on all replacement figures and update the figure caption with source information. If applicable, please specify in the figure caption text when a figure is similar but not identical to the original image and is therefore for illustrative purposes only.

Reviewers' comments:

Reviewer's Responses to Questions

**Comments to the Author**

1. Is the manuscript technically sound, and do the data support the conclusions?

Reviewer #1: Yes

Reviewer #2: No

2. Has the statistical analysis been performed appropriately and rigorously? 

Reviewer #1: Yes

Reviewer #2: No

3. Have the authors made all data underlying the findings in their manuscript fully available?

Reviewer #1: Yes

Reviewer #2: No

4. Is the manuscript presented in an intelligible fashion and written in standard English?

Reviewer #1: Yes

Reviewer #2: Yes

5. Review Comments to the Author

Reviewer #1: My sincere thanks to the editor for the opportunity to review this manuscript.

It has been a pleasure for me to read the manuscript which I believe makes a relevant contribution.

There are many virtues of the manuscript, starting with a topic that is highly current, that changes continuously and that has a profound social relevance. In addition, the authors who carried it out the research are highly recognized and well versed in the subject they address. The paper updates and builds on previous knowledge of the subject, providing current data that can inform different social agents who are concerned (and should be concerned) about this phenomenon.

I simply have a few issues that I think can be improved, but having dealt with them I believe that the paper will be useful for the readers of the journal and that it will make a scientific contribution to the research community.

Here are a few minor issues that I have detected:

‘attempting to ban loot 82 boxes may also be impractical, as demonstrated by how Belgium failed to do so’ seems like a very rigorous statement…perhaps something more subtle as has not been able to tackle or be innefective would be more sensitive

I cannot help but wonder why is the manuscript so heavily focused on Iphone leaving out android. A quick search reveals that according to statista as ‘of June 2022, iOS had 50.99 percent of the market share of mobile operating systems in the United Kingdom (UK), while Android followed closely with over 48 percent’ Given that Android phones in general are cheaper that IPhone I wonder if a SES bias is not in place when only checking Iphones. Moreover I think (I am not an expert on these matters) the work flow in the Play Story (from android) is different form that of the Apple Store, which could lead to very different results in Android phones. Perphas this should be addressed in the limitation section of the paper.

Figure 2 is not very clear in my opinion, could it be made more clear of be explained better in text? (I found this information later in the manuscript on line 377) perhaps the Figure could be place closer to this information)

On page 11 a sentence start at line 320 and end at line 327. Such a long sentence is very hard to follow.

How I would know which are the games that were analysed? How would I know which are games 1 and 26 (page 11, line 320) Could not they be presented in the manuscript as a table or igure?

Why is the section labeled as ‘Confirmatory Analyses’ as confirmatory analysis. Could not be the title a little bit more informative? What is that the authors are trying to confirm, what were the analyses performed? I think that the section titles should be more informative overall

I found odd that the Discussion starts with Hypothesis 2 rather than Hypothesis 1. Even if this confirmed I think that it would be a much more natural way to start the discussion section.

Reviewer #2: What are the odds? Poor compliance with UK loot box probability disclosure industry self-regulation

Title can be reframed like:

What are the odds? UK loot box probability disclosure sector self-regulation noncompliance.

Abstract: I completely dissatisfied the overall abstract. I didn’t find much information on Purpose, need of the study, Methodology, findings and Practical Implications.

Introduction is less and not up to the mark. I didn’t find much information in the introduction section that can convince the readers that your research subject is interesting and that they should continue reading your paper. A detailed background is missing.

Literature review: It is quite dissatisfactory. Please focus on the research gap in the introduction. Strengthen the argument of your study by referring to a larger body of scientific literature, and clearly outline what is missing in the literature and what gap you wish to fill in.

Please rewrite the whole Research methodology

The author has written a small portion of analysis and discussion part in conclusion.

There is a need of division of discussion and conclusion into two separate sections, with the conclusion section simply needing to give the most critical information in points so that readers can find what they want from the publication in a short time. The discussion section might need to add more subheadings, such as discussion of the findings, main contributions, policy recommendations, limitations, and future work of this study

6. PLOS authors have the option to publish the peer review history of their article (what does this mean?). If published, this will include your full peer review and any attached files.

Reviewer #1: No

Reviewer #2: **Yes: **KIRAN SOOD

<quillbot-extension-portal></quillbot-extension-portal>

---

## [Author Response · Author response to Decision Letter 0]

25 Apr 2023

Our point-by-point response is attached to the manuscript submission system.

---

## [Editor Report · Decision Letter 1]

22 May 2023

What are the odds? Poor compliance with UK loot box probability disclosure industry self-regulation

PONE-D-23-02587R1

Dear Dr. Xiao,

We’re pleased to inform you that your manuscript has been judged scientifically suitable for publication and will be formally accepted for publication once it meets all outstanding technical requirements.

Kind regards,

Simon Grima, PhD

Academic Editor

PLOS ONE

Additional Editor Comments (optional):

Reviewers' comments:

<quillbot-extension-portal></quillbot-extension-portal>

---

## [Editor Report · Acceptance letter]

23 May 2023

PONE-D-23-02587R1 

What are the odds? Poor compliance with UK loot box probability disclosure industry self-regulation 

Dear Dr. Xiao:

I'm pleased to inform you that your manuscript has been deemed suitable for publication in PLOS ONE. Congratulations! Your manuscript is now with our production department. 

Kind regards, 

on behalf of

Professor Simon Grima 

Academic Editor

PLOS ONE